# Scoping Review: Methods and Applications of Spatial Transcriptomics in Tumor Research

**DOI:** 10.3390/cancers16173100

**Published:** 2024-09-06

**Authors:** Kacper Maciejewski, Patrycja Czerwinska

**Affiliations:** 1Undergraduate Research Group “Biobase”, Poznan University of Medical Sciences, 61-701 Poznan, Poland; czerwinska.patrycja@ump.edu.pl; 2Department of Cancer Immunology, Poznan University of Medical Sciences, 61-866 Poznan, Poland; 3Department of Diagnostics and Cancer Immunology, Greater Poland Cancer Centre, 61-866 Poznan, Poland

**Keywords:** spatial transcriptomics, cancer biology, data analysis, Visium, Seurat

## Abstract

**Simple Summary:**

Spatial transcriptomics is a technique of measuring gene expression with spatial resolution on a tissue slide, which is especially useful in cancer research. This scoping study reviews 41 articles published by the end of 2023 to examine current trends for employed methods, applications, and data analysis approaches for spatial transcriptomics, identifying challenges and practical uses of this technique in studying cancer. The research shows that cancer is a major focus in spatial transcriptomics studies, with certain tools and methods being particularly popular among scientists. However, many studies do not fully explain their data processing methods, making it hard to reproduce their results. Emphasis on transparent sharing of analysis scripts and tailored single-cell analysis methods for ST is recommended to enhance study reproducibility and reliability in the domain. This work may stand as a spatial transcriptomics developmental snapshot and reference to contemporary neoplasm biology research conducted through this technique.

**Abstract:**

Spatial transcriptomics (ST) examines gene expression within its spatial context on tissue, linking morphology and function. Advances in ST resolution and throughput have led to an increase in scientific interest, notably in cancer research. This scoping study reviews the challenges and practical applications of ST, summarizing current methods, trends, and data analysis techniques for ST in neoplasm research. We analyzed 41 articles published by the end of 2023 alongside public data repositories. The findings indicate cancer biology is an important focus of ST research, with a rising number of studies each year. Visium (10x Genomics, Pleasanton, CA, USA) is the leading ST platform, and SCTransform from Seurat R library is the preferred method for data normalization and integration. Many studies incorporate additional data types like single-cell sequencing and immunohistochemistry. Common ST applications include discovering the composition and function of tumor tissues in the context of their heterogeneity, characterizing the tumor microenvironment, or identifying interactions between cells, including spatial patterns of expression and co-occurrence. However, nearly half of the studies lacked comprehensive data processing protocols, hindering their reproducibility. By recommending greater transparency in sharing analysis methods and adapting single-cell analysis techniques with caution, this review aims to improve the reproducibility and reliability of future studies in cancer research.

## 1. Introduction

Spatial transcriptomics (ST) is a method to capture gene expression data with corresponding spatial context on the tissue slide, which allows the analysis of one of biology’s principles—directly relating tissue morphology with its underlying function, according to the “form follows function” phenomenon. The context of expression location (this is, captured mRNA within ST spots or regions in specific locations on a tissue slide) allows for the uplifting of a biological assumption that similar cell subpopulations are found physically close together [1]. This is especially interesting for cancer research, capturing the full picture of complex tissue heterogeneity. Tumor studies are a big part of current spatial transcriptomics use. Up to the end of 2023, The Gene Expression Omnibus (GEO) lists over 800 unique samples used for cancer research, including normal healthy controls (Figure 1A), with the greatest representation of pancreatic, breast, and liver carcinoma tissues. However, the field of spatial transcriptomics is not a new concept. The first working protocols were published as early as 1987 [2]. Yet, recent advancements directly addressing higher capturing resolution and platform versatility made the method more powerful and thus popular, gaining the title of Nature’s Method of The Year in 2020. Since then, a high uptrend of published manuscripts has been observed, especially for cancer research (Figure 1B).

There are two prominent families of methods for profiling transcriptomes while preserving spatial information: imagining and sequencing ones [2,3,4]. The first category involves imaging mRNAs in situ using microscopy, forming the basis of imaging-based ST technologies. In this approach, mRNA species are distinguished either by (1) hybridizing mRNAs to fluorescently labeled, gene-specific probes (in situ hybridization, ISH, e.g., Nanostring’s CosMx [5], Vizgen’s MERSCOPE [6], or SpatialGenomics’ seqFISH [7]), or by (2) in situ sequencing (ISS) of amplified mRNAs within tissue sections using sequencing by ligation (SBL) technology (e.g., 10x Genomic’s Xenium, based on Cartana [8], and FISSEQ [9]) [10]. The second family of methods involves extracting mRNAs from tissue while maintaining spatial information and then profiling the mRNAs using next-generation sequencing (NGS). This sequencing-based ST relies on preserving spatial information through methods such as microdissection (e.g., Nanostring’s GeoMx [11]), microfluidics (e.g., non-commercial DBiT-seq [12]), or by ligating mRNAs to spatially barcoded probes in a microarray (e.g., 10x Genomics’s Visium—formerly Spatial Transcriptomics [13], STOmics’s Stereo-seq [14], or non-commercial slide-seq [15]) [10].

Array-based methods for sequencing-based ST involve capturing mRNAs from tissue sections using spatially barcoded arrays. These arrays, which are analogous to pixels in imaging, retain positional information at the point of mRNA capture [13]. In Visium technology (10x Genomics), tissue is placed on an array where mRNAs are captured by spatially barcoded probes, converted to cDNA, and then sequenced (Figure 2). The spatial resolution is determined by the size of the capture areas. Despite their advantages, such as the ability to profile large tissue sections and untargeted transcriptome analysis, array-based methods have limitations: they often lack single-cell resolution, as mRNAs from different cells can be captured at the same spot [10]. Microdissection methods for sequencing-based ST involve physically isolating specific regions or cells from tissue sections to preserve spatial information. An early example is laser capture microdissection (LCM), which allows for precise removal of tissue regions for transcriptomic profiling via microarrays [16]. However, LCM can cause mRNA degradation due to laser-induced damage [17]. Modern microdissection techniques, such as Nanostring’s GeoMx, overcome this limitation, using gene-specific photocleavable probes to barcode mRNAs, which are then released by UV light in targeted regions and sequenced (Figure 2) [11].

Among sequencing-based techniques, all mentioned commercial solutions (Visium, Stereo-seq, and GeoMx) offer paired subsidiary staining like H&E, enhancing the range of their possible applications. While all these platforms offer high-throughput capabilities, array-based Stereo-seq excels in achieving single-cell resolution [19]. GeoMx uses a targeted ROI (region of interest) selection strategy within the tissue, allowing precise capture of histologic structures’ gene expression profile. As the ROI area is limited, the Visium and Stereo-seq platforms may be better suited to answer biological questions at the level of the entire tissue section [20]. 10x Genomics has recently developed other commercially available technologies like Visium HD and Xenium. With its enhanced resolution, array-based Visium HD enables finer-scale mapping of gene expression patterns, achieving a single-cell level of the whole tissue [20]. Xenium, as the latest 10x Genomics in situ sequencing ST technology, provides even higher resolution, allowing for the capture of gene expression profiles at a subcellular level. It is reported that in terms of sensitivity and single-cell alignment accuracy, Xenium may outperform its market competitor Nanostring’s CosMx ISH-based platform [21]. It is important to mention that this choice of possible ST platforms is not exhaustive, as there exist other solutions, among them those that are still in the development phase being currently not commercially available, like Slide-seq [15] (Francis Crick Institute, United Kingdom), XYZeq [18] (University of California, USA), or seq-scope [22] (University of Michigan, USA).

ST data analysis methods and applications are heterogeneous in the latest tumor-related literature. Thus, this scoping review aims to map current trends for employed methods, applications, and data analysis approaches for ST in the domain of neoplasm research, also addressing possible challenges a researcher may face while performing ST analysis. As currently any golden standard for ST data analyzes in cancer research is yet not established, the focus is shifted towards software, tools, and data manipulation techniques that are associated with specific applications. Moreover, the quality of data analysis reporting is also assessed. The study builds upon already-published reviews by Fang et al. [23] and Williams et al. [10], emphasizing practical applications in cancer research to help decide whether one’s study may benefit from additional spatial context.

## 2. Materials and Methods

### 2.1. Quantity of Deposited Samples and Studies

Firstly, we searched the GEO (Gene Expression Omnibus) database (https://www.ncbi.nlm.nih.gov/gds/, accessed on 4 February 2024) to enumerate the number of spatial transcriptomics samples of tumor tissues available publicly in this repository. We used the query *(spatial transcriptomics) AND (tumor OR tumour OR neoplasm OR cancer OR carcinoma)* to look for all the tumor-related samples with spatially resolved gene expression. We made a list of all the organs from which the deposited tissues originate and then built up individual queries for each organ-associated neoplasm. The search was restricted to only human samples published up to 31 December 2023. Later, we took the three most commonly identified organ types and performed additional PubMed queries (https://pubmed.ncbi.nlm.nih.gov/, accessed on 4 February 2024) to evaluate the number of published articles (up to 31 December 2023) in general (thus without a priori filtering) that use organ-related samples for ST analysis. The individual search queries for this analysis can be found in Appendix A.

### 2.2. Article Search and Data Charting

To analyze the current trends of spatial transcriptomics applications and employed methods in neoplasm biology research, using a systematic data retrieval approach, we reviewed all original research articles that include structured abstracts published up to 31 December 2023 in the field of ST. This scoping analysis was conducted with the PRISMA-ScR guidelines in mind (Preferred Reporting Items for Systematic reviews and Meta-Analyses extension for Scoping Reviews Statement—https://www.prisma-statement.org/scoping, accessed on 4 February 2024) [24]. Any already-existing protocol was not used, and no protocol was previously registered as a new one for this analysis.

#### 2.2.1. Study Selection

To be included in the review, papers needed to present novel tumor biological findings using ST technology. Thus, the inclusion criteria were as follows: (1) The study utilized tumor samples; (2) the study performed sequencing-based ST data analysis; (3) the study focused on novel biological discoveries. Additionally to wet-lab papers, purely computational studies were also included. We looked for original research articles, including *spatial transcriptomics* combined with either *cancer*, *tumour*, or *tumor* search tags. We narrowed down our search to peer-reviewed studies published up to 31 December 2023, which were written in English and included structured abstracts. Filtering only articles with structured abstracts is a major limitation of our study, as such a requirement is often journal-specific. However, it made our high-throughput systematic analysis more feasible for manual data extraction. We excluded papers that (1) focused entirely on presenting new ST data analysis tools or on comparing different ST methodologies without emphasizing any novel biological insights or (2) used already-analyzed ST data without performing any own ST data processing/analysis/integration. To collect relevant documents, we searched the PubMed database (https://pubmed.ncbi.nlm.nih.gov/, accessed on 8 February 2024). Our search workflow is presented in Figure 3. No duplicates were detected at the stage of full-text reads. We employed the one-query search strategy and only one database to ensure our analysis is fully reproducible and continuable in an easy manner; thus, the search was not supplemented with the scanning of reference lists of relevant reviews. The full query (search strategy) for the PubMed repository is present in Appendix A.

We conducted abstract reads of all 70 articles and filtered out 41 of them as relevant. Among them, studies published by Wang et al., Zeng et al., and Yoosuf et al. [25,26,27] focused on machine-learning applications of ST data, while the rest of the articles were fully committed to exploring novel biological insights. Among rejected articles, we found ones presenting purely utility tools for ST data analysis without any novel discoveries, ones not using tumor tissues, ones that were methodology protocols for sequencing library preparation, or ones unrelated for other reasons. The full list of rejected articles, along with our reasoning, is presented in Appendix A.

#### 2.2.2. Data Retrieval

Firstly, all articles were briefly screened to determine the variables to extract via the data-charting form. The whole data-charting process was performed twice in an independent manner, each time using the same charting form that was being continuously updated in an iterative process. Based on structured abstracts, we identified the main study aim for each study, collected associated keywords for studies whenever available, and classified each article into the organ category of the tissues from which the ST analyses were derived. We performed a full-text read of 41 selected articles to identify (1) general conclusions with a focus on ones obtained from ST data analyses and (2) whether data and code were available whenever applicable (e.g., whenever authors collected and processed their own samples using any programming language-based tools). The methodological section of every article was independently read twice at one-week intervals within each data-charting round to detect possible inconsistencies in collected items. We were manually looking for the following information: ST sequencing-based technology (platform); source of ST samples (whether original ones or already-public ones); all mentioned computational tools, methods, and functions for processing ST data; employed ST data normalization techniques; mentioned ST data dimensionality reduction protocols; reported ST data availability; whether quality control (e.g., filtering, spot removal, and sequencing quality control) was described in detail (allowing exact reproduction of results); and other data types employed by authors alongside ST data. Additionally, based on the methodological section findings, we thoroughly analyzed the results sections of all studies to identify the types of analyses for which ST data were applied. After each round of data-charting, all information was manually filtered and corrected to ensure the same format for each category (thus eliminating all typos, rewriting names of the same synonymous items, etc.) and to ensure all tags were appropriate for summary statistics analyses. If applicable, the definitions of all categorical items within each variable are included in each figure description.

### 2.3. Data Analysis

Summary analyses of all variables were carried out in R 4.3.3 (The R Foundation, Vienna, Austria) with additional ggplot2 (v3.4.2) [28] and ggprism (v1.0.4) [29] libraries as data visualization aids. Details regarding specific study groups and variables used for each plotting are included in each figure description. Appendix A summarize selected variables.

## 3. Results

Figure 3 and Appendix A represent the flow diagrams for the process of selection of evidence sources. Appendix A overviews retrieved articles, including identified platforms, tools and methods employed by their authors to process ST data. A general overview of study aims and ST applications in analyzed articles is collected in Appendix A.

### 3.1. Tumors Types and Availability of Materials

Using search tags related to spatial transcriptomics and tumors in general, we did not find any article published before 2020 that would fall under our selection criteria. Thus, the oldest and the only study published in 2020 was the one conducted by Yoosuf et al. [27], where authors applied ST technology to create a support vector machine model able to distinguish ductal carcinoma in situ from invasive ductal carcinoma regions in clinical biopsies based on spatially resolved gene expression. This proof-of-concept study used publicly available samples sequenced on precursor ST technology, from which Visium (10x Genomics) NGS-based technology later originated [30]. As presented in Figure 4A, most of the studies within our selection criteria were published in 2023. Among studies that collected and analyzed their own samples (samples were therefore not taken from any publicly available repository), almost 70% of studies shared the access numbers and repository names, making it possible to reuse the newly sequenced samples by other scientists (Figure 4B). Unfortunately, we also found studies that did not mention at all whether their data were available in any way. Even more concerning, only 20% of articles that mentioned employing programming language-based tools for ST data analysis shared the analysis code alongside their published results. This is worrisome, as specific implementations of employed tools and functions are heavily user-dependent, and code exclusion leaves room for potential result manipulations or method misuses.

Analyzed articles utilized tissues derived from many organs for their ST protocols, mostly originating from the central nervous system region (including glioblastomas, medulloblastoma, and ependymoma), lymphoma, breast, liver, and pancreas (Figure 4D). The breakdown of detailed studied tumor types in each analyzed article is included in the collective Appendix A.

### 3.2. Sample Source and ST Platforms

We analyzed the sample source of each study. Most authors collected and fully processed their samples from hospital patients undergoing tumor resection (Figure 5A). In a few cases, authors utilized mouse xenografts. Some studies used ST samples from GEO database, utilizing datasets published previously by other authors. This finding highlights the importance of publishing raw data with original articles to allow other scientists to derive new insights from existing data. Three studies utilized data from the scCRLM Atlas (Single-Cell Colorectal Cancer Liver Metastases—http://www.cancerdiversity.asia/scCRLM/, accessed on 5 May 2024), the compendium containing pre-processed gene count matrices for colorectal and liver carcinomas. Last but not least, we found three studies utilizing data from the 10x Genomics database (https://www.10xgenomics.com/datasets, accessed on 5 May 2024), which contains over 70 human ST datasets of 11 different tumors and also includes free healthy controls for reference. 10x Genomics, as a biotechnology manufacturer, offers sample data for all of their ST technologies, including Visium, Visium HD, and Xenium.

As presented in Figure 5B, Visium (10x Genomics) technology was a sequencing-based technique of choice among studies that collected their own samples. Only five articles in our analysis employed other technologies like GeoMx (Nanostring) and Stereo-seq (STOmics) for original sample generation.

### 3.3. Quality Control and Data Normalization

Firstly, we analyzed the amount of necessary details regarding ST data quality control authors report in their studies. This sequencing analysis step includes filtering based on minimum gene counts per spot, minimum sequencing depth, maximum mitochondrial gene expression share, or manual region correction and filtering to e.g., include only sequencing data directly from the tissue and not to count so-called background regions that do not contain tissue fragments underneath. Only 55% of studies reported detailed, reproducible protocols of their quality-control approaches, including filtering thresholds (Figure 6A). Another 30% of studies mentioned that some filtering of low-quality reads was performed but without providing any specific details. Such an approach is insufficient; the filtering and thus spot selection for the analysis is crucial, as it may influence the results by various levels of accepted background noise [31], making the direct comparisons across studies inaccurate and unrepeatable. Around 15% of studies did not report any filtering or quality check details at all, and it is unknown whether they were performed, making their results even more irreproducible. Detailed descriptions of wet-lab experimental methodologies have become a good practice standard in biomedical research. According to our findings, researchers generally pay less attention to describing employed computational data analysis protocols for unknown reasons.

We analyzed the most common approaches for ST data normalization. This step occurs after read trimming, read alignment, and mapping identified transcripts to gene names based on an annotated reference genome (Figure 2). We found out that applied normalization techniques were generally platform-dependent. As presented in Figure 6B, Visium users predominantly applied the *SCTransform* function from the Seurat R package, originally a single-cell analysis toolkit [32]. On the other hand, GeoMx users applied quantile normalization methods (either full or upper quantile ones) implemented in the limma R package which was originally developed for microarray and bulk RNA-seq data analyses [33]. Around one-third of studies did not report any specific normalization method, mentioning only that some unambiguous normalization was performed. This is concerning, as different data normalization approaches may lead to various forms of introduced bias, determining various limits of data interpretation and possible conclusion drawing, especially when dealing with multiple samples from heterogeneous patients [34,35,36].

### 3.4. Application-Oriented Data Analysis Software

Applied methods are strictly related to analysis types, and thus, the employed data analysis software matches the authors’ application purposes for utilizing ST data. To investigate the software used by authors to analyze ST data, we carefully read the methodological sections (including Appendix A, if applicable) and extracted all computational tools mentioned in every analyzed article. We counted the number of times each software was mentioned throughout our analysis. As different ST platforms output sequencing results in different data formats, we split articles based on the utilized platform: either Visium (Figure 7A) or GeoMx (Figure 7B). The only study employing Stereo-seq (STOmisc) technology used Seurat and spacexr [37] R libraries [38]. Note that the mentioned tools (e.g., Seurat) may not necessarily be exclusive to one specific ST platform.

Visible platform-dependent differences in specific software usage may result from different analysis types performed by different authors, utilizing either Visium or GeoMx platform datasets. Seurat, as a major bioinformatic R-based toolkit, initially suited to single-cell sequencing analyses, was commonly used for users of both of these ST platforms. The second most-used software for Visium analyses after Seurat was SpaceRanger, which is a full analysis pipeline designed specifically for Visium data alongside brightfield or fluorescence microscope images, developed by Visium’s manufacturer—10x Genomics. This pipeline enables users to run all necessary sequencing data pre-processing steps like read demultiplexing, read trimming, and read reference genome aligning, including ST-specific steps like mapping the entire transcriptome across tissue slides. The other commonly mentioned software developed by 10x Genomics is LoupeBrowser. This tool allows filtering, clustering, and manual expert annotating aligned with the tissue sequencing regions (through categorizing spots by, e.g., specific tissue types on the slide). On the other hand, the predominantly used AtoMx is analogous to SpaceRanger and LoupeBrowser software developed by GeoMx’s manufacturer, NanoString, to pipeline analyze data generated using this platform. All other mentioned software was developed by third parties.

Even though we observed limited use of single-cell-rooted scanpy, a Python-based Seurat alternative [39], we did not find any use of squidpy, a new scanpy’s ST-tailored forge [40]. However, due to the ease of machine-learning applications and faster computation times, in general, Python has recently gathered more and more attention in the field of data science [41]. We selected the most prominent identified ST-related analytical software alongside the reasons for their application in analyzed studies in Table 1 (besides Seurat library, as it is discussed throughout this whole article). We did not systematically collect some general-purpose data visualization tools, as they were not directly associated with processing ST data. However, authors commonly used ggplot2 [28], ggsignif [42], pheatmap [43], ggstatsplot [44], and complexheatmap [45] R packages and matplotlib [46] and seaborn [47] Python libraries.

### 3.5. Data Integration and Dimensionality Reduction

Further on, we collected all the function names listed in the methodology sections. Not every article reported those, but common trends can be derived from our partial results. As presented in Figure 8A, almost all functions that were mentioned in at least two articles belong to the Seurat R library. Seurat stands as a comprehensive sequencing data analysis toolkit, providing many functions for standard secondary analysis, and as the collection of the most common implementations of widely applied methods [83].

There are two main concepts regarding the integration of multiple samples or data types [84]. In principle, horizontal integration compares the same data type from different samples. In the context of spatial transcriptomics, horizontal integration may also refer to the integration of multiple sections of the same organ, allowing the creation of a 3D visualization and analysis of the organ fragment. On the other hand, vertical integration combines multiple samples originating from the same patient. In general, it may include different heterogeneous data types. We noticed that integrating ST data with single cells is a common phenomenon. Single-cell data can leverage ST data with unique insights regarding, e.g., specific cell types present in the individual spots (namely, cell-type deconvolution). Integration itself in Seurat is achieved through the so-called anchor method, which enables the probabilistic transfer of annotations from a reference to a query set [83]. This protocol includes *FindTransferAnchors* and *TransferData* functions. As an example, Guo et al. utilized the *FindTransferAnchors* function to map each cell type to the spatial region and effectively obtain access to the distribution of various cell subgroups in the spatial context [53]. Alternative solutions for data integration and single-cell/spatial data analysis in R are in general offered in the Harmony library [65]. Two studies also reported another data integration technique through multimodal intersection analysis (MIA), initially proposed by Moncada et al. [85]. As applied by Guo et al., each resulting cluster in the spatial embedding was defined as predominantly abundant with this cell type, whose characteristic genes had the highest enrichment significance of intersection [53]. 

Spot clustering is a technique to group spots/cells based on their expression pattern similarities. We identified nearest-neighbor clustering algorithms to be used the most widely, supposedly because this approach is also implemented in the *FindClusters* function, which was identified as commonly used in our analysis and thus often reported by authors. Other mentioned clustering methods were as follows: Leiden clustering, Ward clustering, Louvain clustering, and hierarchical clustering (which was sometimes reported only as a visualization aid for heatmaps).

High-throughput sequencing technologies like modern spatial transcriptomics solutions generate records for over 30,000 transcripts per one spot (or one cell in the case of single-cell transcriptomics). Thus, there is a need to perform so-called dimensionality reduction to limit interest genes to the most biologically significant ones. As shown in Figure 8B, most analyzed studies employed a combined approach of PCA (principal component analysis) and UMAP (uniform manifold approximation) to select the most important gene expression features or purely for visualization purposes. This method was preferred by authors of studied articles to the PCA method alone.

As presented in Figure 8C, only 3 out of 43 articles based published results only on ST data. Authors eagerly cross-referenced their results with immunohistochemistry methods (including multiplex immunohistochemistry and immunofluorescence), typically to validate ST findings (such as abundance distribution within tissues) on a protein level. Other sequencing types were also commonly used alongside ST, mainly single-cell transcriptomics, which allows, e.g., cell-type deconvolution of ST spots or other integrative analysis. The rest of the utilized omics technologies were standard bulk sequencing and more specialized single-nucleus transcriptomics.

### 3.6. Semantic Analysis and Application Purposes

We performed the analysis of keywords to extract the key research interests of researchers using ST data in the domain of neoplasm biology. We found that besides keywords related to the (1) ST technology itself and the (2) utilized anatomical region, the most common keywords were *tumor microenvironment*, *heterogeneity*, *hypoxia*, *immunotherapy*, and ones related to single-cell sequencing (Figure 9A). This supports our previous findings that single-cell technologies are commonly used alongside ST datasets (Figure 8C and Figure 9A).

We also systematically categorized every analyzed article to understand the most common purposes of applying ST protocols (Figure 9B). Most of the found articles performed analyses related to uncovering tumor landscape composition, tumor microenvironment, tumor heterogeneity, or cell–cell interactions. As included in this review, articles perform different analyses to different extents and in different manners; to reduce the bias of our study as introduced by possible subjective segmentation into more detailed overlapping categories, here, purely for visualization purposes, we classified them all as *tumor heterogeneity*. However, we emphasize that tumor microenvironment, tumor heterogeneity, and cell–cell interactions are all distinct biological phenomena and often can be defined in multiple ways. Here, by *tumor heterogeneity*, we understand the general coexistence of many cellular niches that differ in genotype/phenotype within the tumor(s) and its microenvironment. This covers all definitions of the term itself [86] and may also include interactions between particular niches [87]. In the majority, the usage of this particular term in studies was rather generic and did not implicate any specific common analysis applications. And yet, few researchers have tried to formalize *tumor heterogeneity* by its enumeration and comparison between samples or spots. It could be either by, as implemented by Powell et al., (1) calculating gene expression variability (and thus dispersion) across spots (the higher variability, the higher heterogeneity) or by (2) calculating the Shannon entropy, as applied in the Moffet et al. study [88,89]. Some more robust and ST-tailored approaches have been proposed, e.g., by Levy-Jurgenson et al., who developed the Heterogeneity Average (HTA) index (alongside their Python implementation) for tissue heterogeneity quantification with imagining applications, especially to handle the multivariate nature of ST datasets [90]. Some other studies used ST data purely for simple visualizations of the *expression distribution* of specific target genes, which were selected from their previous findings or the literature. Such studies were not classified into *tumor heterogeneity* due to the lesser complexity and comprehensiveness of this approach. 

In Appendix A, we collected all identified study aims that confirmed researchers’ particular focus on tumor heterogeneity and tumor (immune)-microenvironment.

For researchers, cell–cell interaction analyses could mean specific methodologies with specialized software, as described above (see also Section 3.4), or simpler spot-resolved co-expression analysis. As an example of a more complex cell–cell interaction analysis, Liu et al. employed the previously described NicheNet [72] to infer the mechanisms of interaction between SPP1+ macrophages and cancer-associated fibroblasts (CAFs). They identified a tumor immune barrier structure in hepatocellular carcinoma tissues, which was characterized as a spatial niche composed of SPP1+ macrophages and CAFs located near the tumor boundary that influences the efficacy of immune checkpoint blockade [49].

Gene set enrichment analyses (GSEA) [91] were commonly performed to analyze pathways or specific signatures enrichment, including the detection of enriched pathways and molecular processes, gene ontology, immune activity, and methodologies like GO and KEGG analyses [92,93]. 

Cell-type deconvolution was always performed with the support of annotated single-cell data to estimate immune cell infiltrations or identify specific cancer-related cells. For instance, Hong et al. employed MCP-counter [94] to map and identify immune cell populations [95]. However, as presented in Section 3.4, a wide selection of various tools for cell-type analyses is available. 

Weighted correlation network analysis [96] was performed in four articles, although even more studies examined gene co-expression patterns by simpler correlation tests.

### 3.7. ST-Derived Biomedical Advancements

To point out possible capabilities of ST analysis, in this section, we describe selected advancements in neoplasm biology presented in analyzed studies, focusing on the three most studied organs from which tumor samples were collected (Figure 4D). Some of the objectives were achieved with the aid of complementary methods to ST. However, the application of ST plays a fundamental role in all of those discoveries either as a primary source of knowledge or as a complementary element of a bigger picture. We underline that the list advancements is not exhaustive in the context of the whole domain. Information regarding discovery in other tumor types is collected in Appendix A.

#### 3.7.1. Central Nervous System

Heming et al. characterized the malignant B-cell intratumor heterogeneity and T-cell exhaustion in primary central nervous system lymphoma, highlighting the need and potential for personalized treatments [97]. They found spatial proximity of T-cell transcripts to the B cells and further confirmed these results with immunohistochemical staining. Later on, they integrated their Visium-based ST results with generated single-cell sequencing data through the so-called anchor-based method in Seurat [32]. Based on canonical markers, they discovered frequent T-cell exhaustion and their co-localization with proliferation-promoting malignant B-cells clusters, which was previously defined through their single-cell sequencing results. This underlines the comprehensiveness of applying both spatial and single-cell sequencing protocols within the same study.

Vo et al. determined the spatial organization of cellular states in medulloblastoma, including their impact on therapy resistance [63]. Visium assay was performed on mouse xenograft samples; thus, the workflow needed to be adjusted for species-specific tissue stratification within each sample, based on the predominant expression of human or mouse transcripts per each spot. To identify DEGs (differentially expressed genes) between Palbociclib-treated (CDK4/6 inhibitor) and untreated samples, a voom pipeline from limma [33] was fed with pseudobulked ST data (defined as summed UMI counts for each gene across all spots) for both species’ samples individually. Later, they utilized clusterProfiler [50] to perform GSEA on human DEGs, showing upregulation of neuronal differentiation and downregulation of the cell cycle. Gene set enrichment scores per spot were estimated using PAGE workflow from Giotto [62] to depict spatial localization of neuronal differentiation, progenitors, and cell cycle in response to the therapy. Finally, spots were annotated by dominant cell type against single-cell reference using SingleR [80], localizing astrocytes to the mixed tumor microenvironment interface. This work is particularly interesting, as the authors also statistically compared transcriptional heterogeneity between treated and untreated samples, presenting lower heterogeneity levels (quantified with the Shannon entropy and the Simpson index) within treated tumors. Collectively, they discovered a regional response to Palbociclib with reduced tumor cell growth and a promoted switch towards a more mature, neuron-like state. This study highlights the power of ST to fully characterize the tumor response to targeted therapies, offering valuable insights into potential resistance mechanisms.

The study of Fu et al. aimed to understand the tumor microenvironment of childhood brain tumor ependymoma, presenting epithelial and mesenchymal subpopulations and cell transition stages [66]. The authors inferred abundant cell types in clustered Visium-generated ST data by (1) manually inspecting markers within clusters, (2) utilizing the Jaccard index of marker genes overlapping between ST clusters and single-cell data, and (3) ontological analysis. Seurat [32] was used to infer cell cycle phases within each spot, and Slingshot package [82] was utilized to predict pseudotime trajectories. The authors revealed a cellular hierarchy initiating at an identified proliferative progenitor subpopulation in tumor epithelial zones, which was associated with tumor progression and may possibly serve as a novel target for therapeutic development. Myeloid cell interactions, confirmed as the leading cause of epithelial–mesenchymal transition, occurred in regions spatially distinct from hypoxia-induced mesenchymal transition, collectively demonstrating the utility of ST in revealing a portrait of the cellular composition, their interactions, and their influence on tumor progression.

#### 3.7.2. Breast

Tashireva et al. characterized the morphological and spatial heterogeneity of primary breast tumors in the context of expression profiles of integrins and their ligands [71]. They employed the NICHES (Niche Interactions and Communication Heterogeneity in Extracellular Signaling) [70] package on generated Visium data to infer ligand–receptor interactions, followed up by UMAP and SNN clustering to group spots with similar signaling microenvironments. Finally, Seurat’s [32] *FindAllMarkers* function revealed the most significant ligand–receptor interactions within clusters. They discovered spatially heterogeneous integrin–ligand clusters that contribute to breast tumor functional diversity, influencing parenchymal–stromal interactions and potentially promoting metastasis. This intra- and inter-tumoral spatial–functional heterogeneity challenges the therapeutic targeting of single molecules or pathways. This phenomenon was also confirmed by Powell et al. in their study on pharmacogenes that can impact drug distribution and efficacy throughout the tumor [89]. They revealed that their heterogeneous expression, particularly associated with ROS (reactive oxygen species) handling and detoxification, may underlie internal tumor chemoresistance, as the discovered heterogeneity (quantified here as an interquartile range of read counts) was not specific to chemotherapy-treated samples or cell types.

#### 3.7.3. Liver

ST Visium-based analysis by Zhang et al. of neoadjuvant cabozantinib-nivolumab (multi-tyrosine kinase inhibitors) therapy identified independent mechanisms of resistance and recurrence in advanced hepatocellular carcinoma (HCC) [60]. Firstly, they assigned cell types to Leiden clusters based on HVGs (highly variable genes). Later, they ran DEA between responders and non-responders using pseudo-bulked tumor spot counts. GSEA was performed using genes ranked by their fold-change, defined by previous DEA results. To determine the intercellular interaction-induced molecular pathways, cell–cell signaling networks were created through Domino [59] software, using SCENIC-based [74] gene regulatory module activity results. Compared to non-responders, responding tumors displayed enriched pro-inflammatory immune cells and CAF populations within the tumor microenvironment. Results were further validated using CIBERSORT-deconvoluted bulk-seq data. Notably, non-responding tumors showed distinct immune-depleted regions characterized by HCC–CAF interactions and cancer stem cell marker expression. Their ST-driven findings suggest these regions may contribute to early immune escape and tumor recurrence.

Zhao et al. created the spatial map of HCC transcriptomes [98]. The results of t-SNE clustering of Visium-based data indicated that expression profiles were similar between peritumor tissues but different between peritumor and tumor tissues, suggesting highly pronounced intratumor heterogeneity. The clustering of data was further used to identify DEGs based on gene set enrichment in individual clusters. To study involved signaling pathways, the authors employed KEGG [92] and GO [93] analyses, identifying marker genes. Cluster annotation was carried out through correlation of cluster-specific genes with marker genes in each cluster. Collectively, ST analysis identified novel markers for satellite nodule prediction and a six-gene tumor cluster signature for HCC bulk-seq-based prognosis. The authors further validated the clinical significance of marker genes associated with spatial tumor clusters using the Kaplan–Meier methodology. This work clarified intratumoral heterogeneity in HCC, potentially revealing pathogenic mechanisms and novel therapeutic targets.

Another study further explored the immune microenvironment landscape in HCC. Yang et al. discovered that NDRG1 (N-myc Downstream Regulated Gene 1) may contribute to poor prognosis by affecting macrophage differentiation, thus allowing HCC cells to evade the immune system [76]. Here, the use of ST was rather supportive of the findings acquired through other methods. Spacexr [27] and SPOTlight [99] were utilized to perform spatial deconvolution and identify cell types within each spot. The NDRG1+ macrophage cluster was located in the tumor center, indicating the tendency of those cells to chemotaxis, which supported the authors’ previous findings. Later, stLearn [79] was used to infer cell–cell interactions and identify close communication patterns between stem cells, NDRG1+ macrophages, and hepatocytes. The authors also underlined the potential of disulfidosis level (which is a new form of programmed cell death characterized by the disulfide-induced collapse of cytoskeleton proteins and F-actin [100]) in predicting the patient prognosis.

## 4. Discussion

We thoroughly identified current trends in spatial transcriptomics of tumors. We systematically reviewed and described employed data analysis methods, research aims, ST application purposes, and ST-derived conclusions of recent studies that utilized ST tumor samples. It is important to underline that the mentioned methods and applications may not be exclusive to the neoplasm research domain, and many of them are also utilized in other fields like developmental biology and neuroscience. This especially concerns the creation of 3-dimensional molecular maps, spatial cell-type deconvolutions, and cell–cell interactions analyses [101,102]. The exclusive selection of articles that include structured abstracts is a major limitation of our study.

We noticed that currently employed computational methods for ST data analysis are relatively heterogeneous in the literature (Figure 6, Figure 7 and Figure 8), and we recommend that a golden standard for ST data analysis, including its comprehensive discussion and reasoning, should be established shortly to ensure that cancer ST research is clinically relatable and with a minimal level of bias. The knowledge synthesis and study comparison is possible only if the methodologies of individual studies are well reported. As we observed, only 20% of articles that mentioned employing programming language-based tools for ST data analysis shared the analysis code alongside their published article (Figure 4C). Furthermore, around one-third of studies did not report any specific normalization method (Figure 6B), and around 15% of studies did not report any filtering or data quality checks (Figure 6A). As we analyzed only recent studies published mostly in 2023, this work is yet another proof of the growing reproducibility crisis in science as well as in the domains of ST and cancer research [103].

As high-throughput spatial transcriptomics techniques are relatively new, most protocols for spatial data employ techniques developed primarily for single-cell sequencing. Normalization of bulk sequencing datasets often starts with library size normalization. This accounts for different sequencing depths (representing the resolution of how many times a library in question was resequenced), allowing for direct comparison of multiple samples in separate libraries [104]. However, similarly to the normalization process in bulk RNA-seq, using library size as a scaling factor in single-cell/spatial analysis may introduce bias towards highly expressed transcripts. This bias is exacerbated by the prevalence of excessive zeros in single-cell/spatial data, compromising the accuracy of scaling factor calculations [105]. Recent reports suggest that individual tissue regions are strongly associated with library size, even after accounting for cell density in each region. As cell density determines the variation of total read counts, total counts per region may be biologically informative [35]. Studies recommend directly applying specialized techniques such as *SCTransform* (proposed in Seurat [32], initially for single-cell data) or SME (proposed in stLearn [99], directly for ST data) that depend on the library size variation across the individual sample [10]. However, Bhuva et al. suggests that as the size of the library is linked to tissue structure, normalizing these effects using commonly applied single-cell RNA sequencing normalization methods (like SCTransform) can adversely impact the identification of spatial domains [106]. Another study on the impact of different normalization techniques on differential expression analysis showed that not normalizing the ST data may be better than library-size correction for clustering tasks, especially with data from spatial technologies with a subcellular resolution like Xenium [107]. Even though more research on this topic should be carried out, preliminary findings underscore the importance of user caution when integrating concepts and tools from single-cell analysis into spatial datasets, as the assumptions underlying these methods may not hold for ST. 

The SME normalization method stands out due to its comprehensive integration of spatial information, morphology, and gene expression data, which collectively offer a more holistic understanding of cellular dynamics within tissues. By leveraging this multidimensional approach, SME effectively captures cells’ complex spatial organization and interactions. Furthermore, it was designed for ST data specifically. To infer morphological distances between spots, this method employs a convolutional neural network with pre-trained weights learned on the ImageNet challenge to enumerate image features, uplifting so-called knowledge transfer strategy [99]. ImageNet collection consists of millions of images of different objects, people, and animals to aid in training general-purpose computer vision models [108]. Although the ResNet50 pre-trained model (as employed in the SME) excels in content identification of general images, comparative analyses conclude that the histopathology-specific pre-trained models (like KimiaNet) are more effective in extracting image features of tissue morphology, resembling an alternative to general-purpose pre-trained models [109,110]. In our analysis, only two studies performed any computational image analysis. Zeng et al. employed *CTransPath*, a transformer-based unsupervised contrastive learning approach trained on massively unlabeled histopathological images, serving as a general tissue feature extractor for histological images [26,111]. Alsaleh et al. took a different approach by utilizing topological data analysis [110]. This neural-network-free and relatively newer analytical paradigm focuses on extracting the underlying structure of data shape, employing mathematical techniques from topology to understand the global and local geometric features of datasets. This method has been proven effective in a histological context and has even been refined further [112,113,114].

Batch effects correction is another challenge of data normalization protocols. This phenomenon arises from differences between samples that are not planned in the experimental design, which can be caused by, e.g., various sample sources, different handlers, experiment locations, or even different batches of reagents [115]. Even two runs of the very same sequencer at different time points can result in a batch effect [116]. Moffet et al. employed the *geomxBatchCorrection* method from the standR R package [117] on GeoMx data to minimize the influence of such an effect over their results [88]. Visium users aiming to address batch effect employed other methods like (1) canonical correlation analysis [118], (2) mutual nearest-neighbors algorithm [69]; (3) so-called anchors method from Seurat R package [119]; or (4) Harmony R package [60,67], which integrates information about cell similarity and batch labels to adjust the gene expression onto a common scale [65]. The need to apply a batch correction can be simply identified by the dataset visualization in less-dimensional embedding using PCA (principal component analysis) or t-SNE (t-distributed stochastic neighbor) [120]. As reported by Liu et al., performed t-SNE clustering showed a significant distinction between samples of spot groups in each sample slice, and thus, they applied the mutual nearest-neighbors algorithm for their batch effect correction protocol [69]. However, for the detection of batch effects, more sophisticated methods are recommended, as simple visual inspections may lead to the risk of unintentionally correcting biologically relevant factors wrongly classified as batch effects [120]. 

Due to recent controversies regarding the strong bias introduced by PCA [121,122], which has been so far broadly used in single-cell workflows [105], it may be suggested to choose a method that does not employ PCA as a dimensionality reduction approach for ST data analysis. PCA has been proven to tend to the selection of U-shaped variables, enhancing patterns that do not necessarily exist in the original dataset [121]. There are multiple other methods of dimensionality reduction, among them non-linear approaches like UMAP (uniform manifold approximation) and t-SNE, which are generally good at summarizing complex high-dimensional data without requiring multivariate normality [123,124]. As the number of published dimensionality reduction methods grows, it is recommended to perform benchmarking internal quality control for the algorithms in question to choose the most relevant one to the dataset. Whichever approach a researcher chooses, it is crucial to be aware of the selected method’s pitfalls and the potential need for algorithm optimization for specific datasets [125].

In our review we found three studies that utilized ST data to aid the development of machine learning (ML) predictive models. For example, Yoosuf et al. found that ST signatures derived from expertly annotated breast cancer tissue samples can effectively identify breast cancer regions in entire tissue sections, including an accurate classification of cancer regions in tissue sections that were not part of the training set [27]. The ST-trained ML model proposed by Zeng et al. applied on hepatocellular carcinoma digital slides was able to estimate progression-free survival in patients treated with atezolizumab-bevacizumab [26]. Generated heat maps indicated which areas are potentially associated with the response to the therapy, offering potential future aid for clinicians. Recent deep learning advancements even allow researchers to accurately predict the expression of hundreds of genes just from a breast cancer H&E tissue microscopy image [126]. ST is a promising technology for such applications, as future integrative image prediction models trained on ST data may be able to make highly accurate predictions based on just histopathology slides, which are currently used nonetheless in pathomorphology as a foundation of tumor diagnosis [126]. This all opens up the exciting possibility for the successful development of novel next-generation pathology diagnostic tools in the future.

## 5. Conclusions

Our results may stand as an ST developmental snapshot and comprehensive reference to contemporary neoplasm biology advancements achieved through ST, including the most popular data analysis approaches and particular tools for specific implementations. Our gathered data may help researchers unleash the full potential of their prospective ST analyzes in future studies. Following up, an extended systematic review of ST methods and applications in neoplasm biology would be beneficial in establishing the gold standards of ST analyses in biomedical practice. Our main conclusions are as follows:Neoplasm biology comprises a great share of ST-based research. The trend is highly increasing, with more and more articles published every year. This also concerns the development of novel data analysis methods and publicly available datasets;Visium (10x Genomics) was identified as the most popular NGS-based ST platform for new sample generation, alongside Seurat’s *SCTransform* normalization and integration method. Studies employing ST analyzes almost always use other data types, such as single-cell sequencing and immunohistochemistry, to support their findings;More attention should be paid to the direct application of single-cell data analysis methods in ST protocols;As all the analyses employing programming language-based tools are becoming more and more complex, authors should publish the analysis code far more often for higher transparency. Moreover, data processing protocols should be described in a fully exhaustive manner, making it possible to exactly reproduce the results;ST technology is predominantly applied to uncover tumor landscape composition and heterogeneity, determine tumor micro-environment characteristics, or identify cell–cell interactions, including spatial expression and co-occurrence patterns. The most common ST application purposes are differential expression analysis, gene set enrichment analysis, spatial cell-type deconvolution, and pathway analyses.

## Figures and Tables

**Figure 1 cancers-16-03100-f001:**
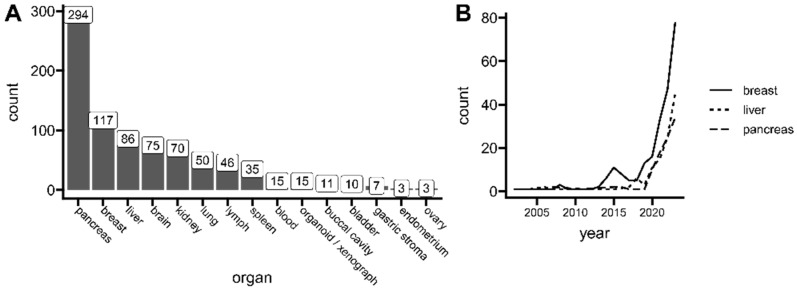
Current interests of spatial transcriptomics in cancer research. (**A**) The number of unique spatial transcriptomics (ST) samples related to cancer research deposited at the Gene Expression Omnibus (GEO) repository. (**B**) The number of spatial transcriptomics (ST) publications related to tumor research per year indexed in PubMed. Data up to 31 December 2023. See the individual queries in Appendix A.

**Figure 2 cancers-16-03100-f002:**
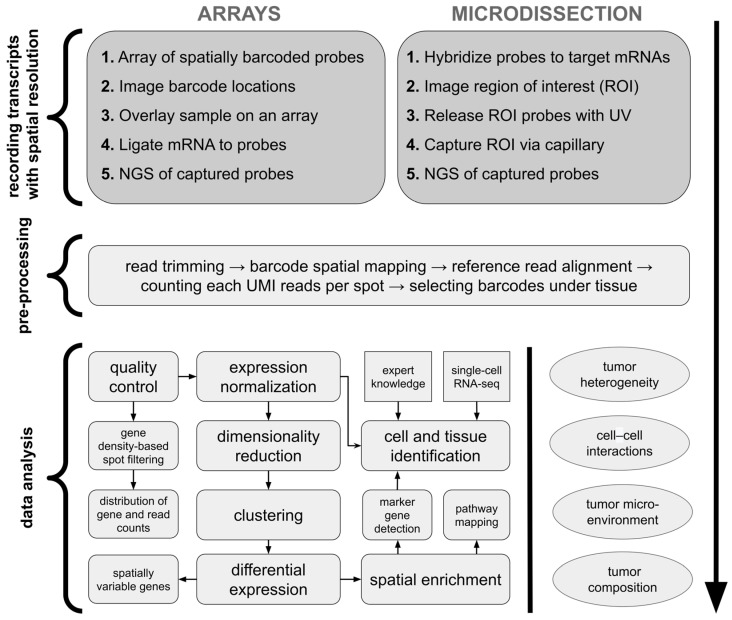
Standard ST analysis workflow of sequencing-based methods. For array-based platforms, mRNAs are labeled with arrays of spatially barcoded probes with a sequence indicating location before sequencing. In the microdissection-based approach, cells or regions of interest can be directly microdissected and their locations recorded before their transcriptomes are sequenced. Later on, sequencing results undergo standardized pre-processing to obtain count matrices including expression values of detected genes per each spot/cell. Next, typical data analysis incorporates quality control with filtering of low-quality spatial regions, expression normalization, feature selection, possible data integration with other samples or data types, tissue segmentation with image analysis, and data annotation. The final steps include data exploratory methods and interpretation, which are highly study-specific; thus, analysis workflows may differ per article. *ROI*—region of interest. *UMI*—unique molecular identifier. Based on Williams et al. [10] and Fang et al. [18].

**Figure 3 cancers-16-03100-f003:**
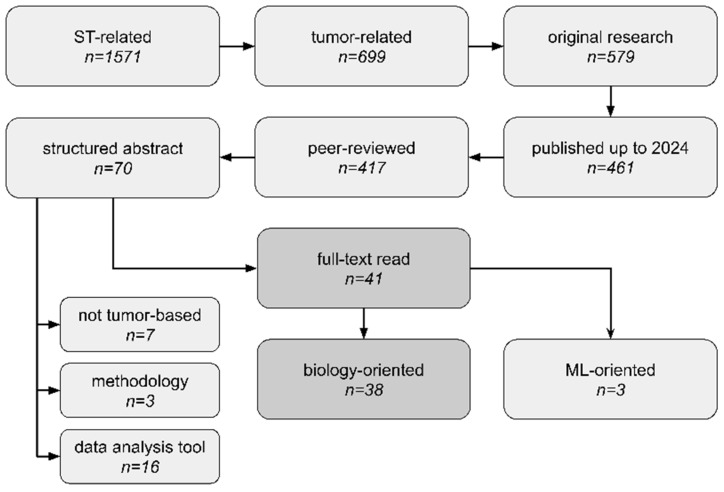
Scoping analysis workflow, study selection criteria, and number of studies at each phase. We thoroughly analyzed 41 articles, which consisted of 38 neoplasm biology-based articles (*biology-oriented*) and 3 articles focused entirely on machine learning (ML) applications in the biomedical domain (*ML-oriented*). *Not tumor-based*—articles did not use tumor tissues. *data analysis tool*—articles focused entirely on presenting new data analysis methods and did not directly derive any new biological insights. *methodology*—articles were related to library preparation or sequencing protocols (e.g., methods comparison).

**Figure 4 cancers-16-03100-f004:**
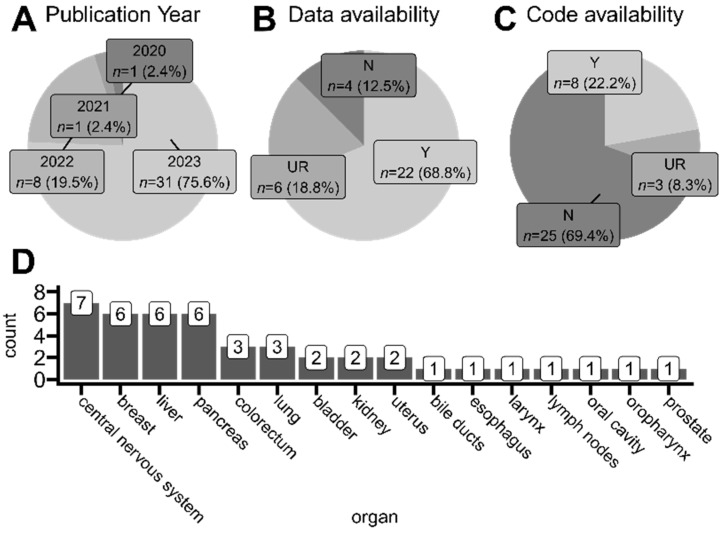
Overview of studies taken into account for the scoping analysis. (**A**) Proportions of articles’ publication years. (**B**) Proportions of data availability approaches of analyzed studies. Only studies that collected and processed their own ST samples are included. (**C**) Proportions of the availability of code developed by analyzed studies. Only studies that used programming language-based solutions (e.g., R/Python libraries) for data analysis are included. (**D**) The number of organs whose sample tissues were collected from for every analyzed study. Two studies utilized samples from multiple anatomical regions. *Y* (yes)—available, authors state the repository and access number where the developed data/code can be found. *N* (no)—unavailable, authors do not mention at all that developed data/code is accessible in any way; *UR* (upon request)—authors declare that the data/code is available upon reasonable request to appropriate corresponding authors.

**Figure 5 cancers-16-03100-f005:**
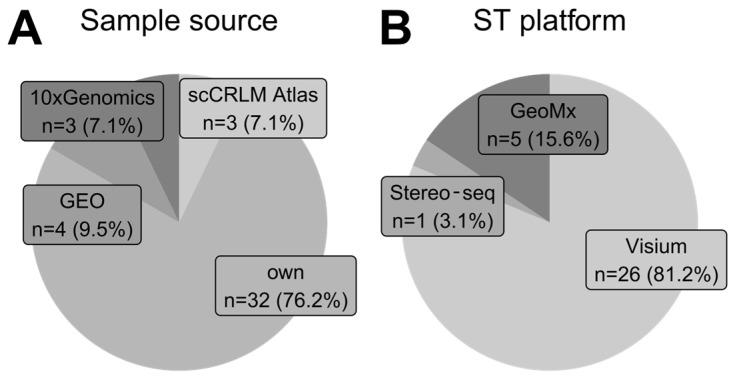
**Trends in sample origin and their development platform.** (**A**) Proportions of sample sources for each analyzed study. The tag *own* represents that authors collected and processed samples fully on their own. Other data sources are publicly available repositories. One study was classified as both *own* and *GEO* because original data findings were later confirmed using publicly available data from GEO. (**B**) Proportions of spatial transcriptomics sequencing-based platforms employed by studies that collected and processed their own samples.

**Figure 6 cancers-16-03100-f006:**
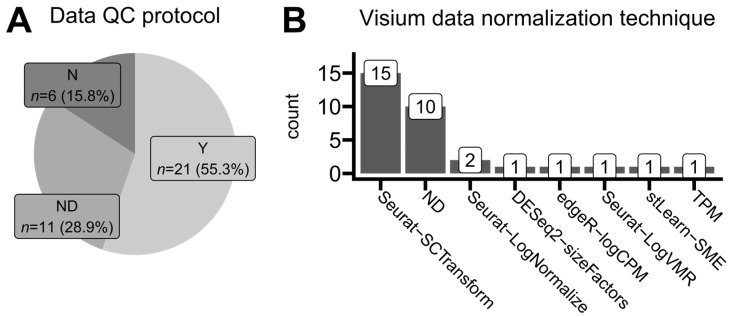
**Trends in standards of data quality-control reporting and expression normalization methods.** (**A**) Proportions of reported data quality-control protocols. Studies analyzing data from the publicly available 10x Genomics repository are not included. *Y* (yes)—quality control (e.g., filtering, removal of specific spots, and sequencing quality) was reported completely and exhaustively with all necessary details. *ND* (no details)—quality control was reported as an employed data processing step but without any explicit information on how it was performed. *N* (no)—authors did not report any quality-control measures in their study. (**B**) The number of times each normalization protocol was mentioned in the analyzed articles that used the Visium (10x Genomics) platform for their spatial transcriptomics data development. One article was classified thrice. *ND* (no details)—authors declared that data were normalized but did not provide any specific algorithm or protocol.

**Figure 7 cancers-16-03100-f007:**
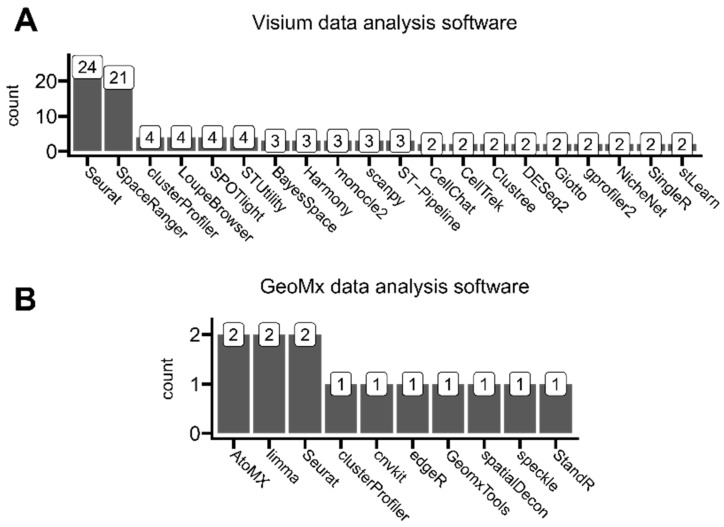
Overview of reported software used for ST data analysis, which was mentioned in (**A**) at least two Visium-based analyzed studies and (**B**) at least one GeoMx-based analyzed study.

**Figure 8 cancers-16-03100-f008:**
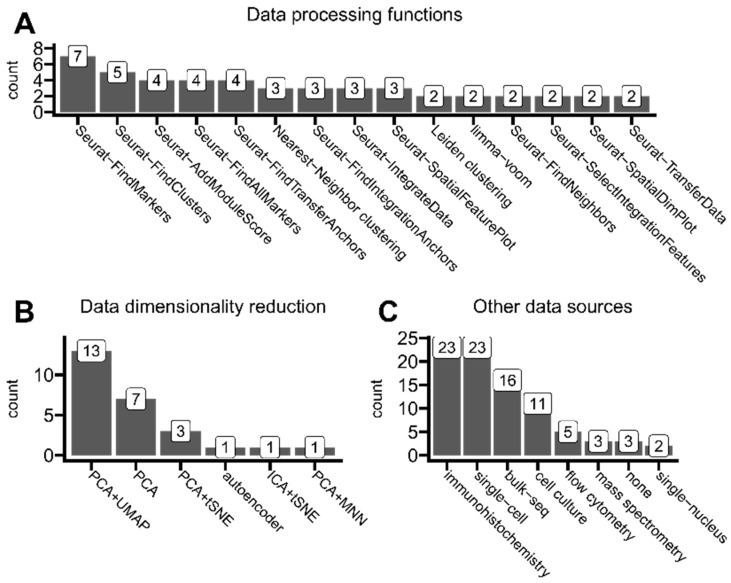
Insights of ST data analysis trends. (**A**) Functions/methods used for ST data processing and analysis (e.g., transforming, reshaping, clustering, and integrating) that were used in at least two articles. (**B**) Data dimensionality reduction methods reported in the analyzed articles whenever mentioned. (**C**) Other data sources used alongside spatial transcriptomics in analyzed articles. The tag *cell culture* represents the group of standard methods and analyses used while performing cell cultures. *Mass spectrometry* also includes mass cytometry protocols. Any animal assays were omitted in this analysis.

**Figure 9 cancers-16-03100-f009:**
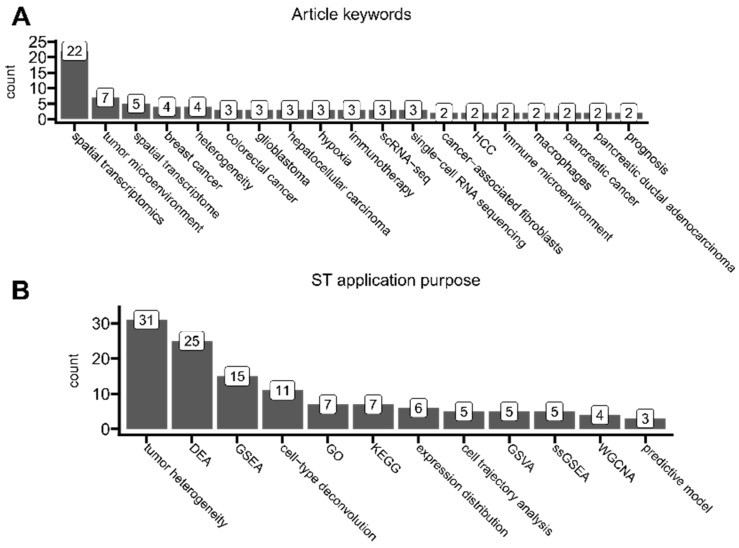
The analysis of current ST application trends. (**A**) Article keywords that were associated with at least two articles. (**B**) Number of times each category of analysis was performed on ST data in at least two articles. Categories were assigned to each article based on a scoping analysis of all articles. *Tumor heterogeneity*—here, an overloaded umbrella term for distinct analyses of tumor landscape composition, tumor microenvironment, and cell–cell interactions, categorized together only for bias reduction of current study. *DEA*—differential expression analysis (e.g., identification of differentially expressed genes between tissue types, highly variable genes, and spatially variable genes). *GSEA*—gene set enrichment analysis (e.g., tumor hallmarks, pathway enrichment, and signature assessments). *cell-type deconvolution*—predictions of specific cell types present in each spot (e.g., immune infiltration based on annotated single-cell data). *GO*—gene ontology analysis. *KEGG*—pathway analysis based on the Kyoto Encyclopedia of Genes and Genomes database. *expression distribution*—simple plotting of spatial expression distribution of genes of interest (this category does not overlap with *tumor heterogeneity*). *cell trajectory analysis*—pseudotime analysis (e.g., spot-wise relative distances). *GSVA*—gene set variation analysis. *ssGSEA*—single-sample gene set enrichment analysis. *WGCNA*—weighted correlation network analysis (e.g., for gene co-expression). *predictive model*—ST data were fed into a machine-learning model to create spatially resolved-bsed predictions.

**Table 1 cancers-16-03100-t001:** Overview of selected unique ST data analysis software.

Software	Language	Examples of Applications
BayesSpace [48]	R	Utilized by Liu et al. to better exhibit spatial expression of features. The spots were enhanced using the *spatialEnhance* function, and the expression features were enhanced with the *enhanceFeatures* function [49].
Cell2location [50]	Python	Utilized by Du et al. to run the deconvolution of spatial tissue locations. To identify the spatial co-occurrence patterns of different cell types, they performed non-negative matrix factorization (NMF) of cell type abundance estimates [51].
CellChat [52]	R	In general, this tool is used to map cellular interactions. It was applied by Liu et al. to evaluate the interaction weights of identified spatial clusters [49]. Guo et al. performed cell communication analysis. To create cell communication networks, they applied CellChat’s functions such as *createCellChat* function, *computeCommunProb*, *computeCommunProbPathway*, and *aggregateNet* [53]. Liu et al. inferred the cell–cell interactome by assessing the gene expression of ligand–receptor pairs across cell types in bulk-seq and ST [54].
CellTrek [55]	R	Employed by Liu et al. to investigate spatial transcriptomics at a single-cell resolution. CellTrek directly mapped cells back to their spatial location by co-embedding the same-tissue single-cell transcriptomics and ST datasets. This approach allowed them to identify malignant cell clusters directly in ST data [54].
clusterProfiler [56]	R	Utilized by Al-Holou et al. to perform gene ontology enrichment analysis, aiming to identify enriched processes amongst the previously identified differentially up- and down-regulated genes [57]. A similar approach was taken by Ren et al. [58].
Domino [59]	R	Used by Zhang et al. to analyze the signaling networks based on the gene regulatory module activities, quantified for each spot by the AUCell/SCENIC tool [60].
GeneSwitches [61]	R	Applied by Liu et al. to identify the order in which functional events are acquired or lost during the transition of malignant cells by processing single-cell data together with pseudotime trajectories to order pathways along the pseudotime. They filtered genes for pathway analysis using the *filter switchgenes* functionality and used the *find switch pathways* module to find significantly changed pathways with pseudotime trajectories [54].
Giotto [62]	R, Python	Vo et al. applied the *PAGE* function, which calculates an enrichment score based on the fold change of cell type marker genes for each spot [62,63]. Cell–cell spatial communication and interactions are available through *cellProximityEnrichment* and *spatCellCellcom* functions and were utilized by Shi et al. [64].
Harmony [65]	R	Employed by, among many, Fu et al. and Yousuf et al. to, e.g., correct batch effects among samples through dataset merging for samples-integrated analysis [66,67].
Monocle2 [68]	R	Liu et al. utilized this tool to simulate the dynamics of temporal development for cell trajectory analysis (aka pseudotime trajectory analysis) by using the spot-resolved expression patterns of previously identified key genes [69].
NICHES [70]	R	NICHES (Niche Interactions and Communication Heterogeneity in Extracellular Signaling) library was applied by Tashireva et al. to assess ligand–receptor interaction between neighboring tissue spots [71].
NicheNet [72]	R	Used by Zhi et al. to access cell–cell interactions between specific cell populations [64].
ReactomeGSA [73]	R, web	Used to score the Hallmark gene set in spatial spots after immune cells scoring through the CIBERSORT algorithm for each spot in the Guo et al. study [53].
SCENIC [74]	R, Python	Employed by Yousuf et al. to perform inference of regulatory modules analysis between transcription factors and downstream regulated genes [67].
scMetabolism [75]	R	Utilized by Yang et al. to evaluate the metabolic activity of each spot [76].
spacerx [37]	R	Used to conduct spatial deconvolution in the Yang et al. study [76].
SPATA2 [77]	R	Used by Andrieux et al. to infer copy number variation (CNV), using the *InferCNV* function. As a result, gains or loss of chromosomes were identified [78].
SPOTLight [79]	R	Utilized by Yang et al. and Liu et al. to identify cell types at spatial spots of interest after their deconvolution [76]. To infer the cell composition of each spot, SPOTLight combined single-cell data and cell-type marker gene information with ST spots and performed a deconvolution based on a NMF decomposition [69].
SingleR [80]	R	Utilized by Vo et al. to map a dominant reference cell type to each spot based on Spearman’s correlation coefficient and nearest label classification [63].
Slingshot [81]	R	Used to infer pseudotime trajectories in the Vo et al. study [63].
SpotClean [82]	R	Applied by Yousuf et al. to remove ambient RNA present in their data [67].

## Data Availability

All complete charted data in a raw CSV format and R scripts for the reproduction of all summary statistics are available upon request from the corresponding author.

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
