# Peer review of "Scoping Review: Methods and Applications of Spatial Transcriptomics in Tumor Research"

_cancers, 2024, doi:10.3390/cancers16173100_

Round 1
Reviewer 1 Report
Comments and Suggestions for Authors
Review:
Spatial Transcriptomics of Tumors: Methods and Applications
The Review focusses on spatial transriptomics on neplasmic human samples. In total 41 manuscript met the inclusion criteria. Thereby considerable ‘laisser-faire’ was discovered in the methodological part concerning data processing was discovered. As the method is quite novel to the field and neoplastic samples yield various hallmarks, this review is very valuable to all (new) users of spatial transcriptomics as well as very helpful to determine rules for maintaining good scientific practices.
However, the authors state the aim of reproducibility, that is very important when analysing the same data sets and highlights the importance of transparent data processing. Although reproducibility in spatial transciptomics, analysing different patient cohorts appears to be almost not possible at the moment. Nevertheless, securing open data processing might be one – very important- variable to determine master regulators.
The authors identify that 36 publications do not share their analysis code, indicating that the manuscript reviewers or the journals do not acknowledge the importance of transparency in their publications. Whether the reviewers of these manuscripts do understand the method or have used the method themselves might be another topic- not addressed in the present manuscript.
Still, this point underlines the importance of this review, as most of the data is published just recently (2020-2024).
Points to consider:
Major:
1. Figure 2 does list the major aspects of the method. However, as the authors try to explain the method to scientists which are not familiar with some of the methods (handling patient tissue, or data processing, etc.) visualisation of the major differences would be helpful. Especially the data processing can be visualized in more detail, especially as the authors found that here most researchers only report poorly their methods- visualisation of its importance, could help researchers to overcome this handicap. Referring to Fang et al- might also be helpful, as they describe it in detail.
2. Line 392, Figure 8C it was explained that 41 articles were analysed, where do the two additional articles come from- described here (43 articles)?
3. The conclusion need to be more concise.
Typos:
line 235: UR instead of UP
line 483: CXCL instead of CVCL
Reviewer 2 Report
Comments and Suggestions for Authors
In the paper submitted to Cancers MDPI titled: Spatial Transcriptomics of Tumors: Methods and Applications by Kacper Maciejewski and Patrycja Czerwinska, authors talk about Spatial Transcriptomics sequencing platforms, methods and Data analysis and applications used in tumors in collection of 41 total scientific papers published in 2023.
The review paper is of interest for publication in Cancers MDPI, though after major revisions. The main critique is lack of well researched example(s) of ST use in cancer research, with in depth given description of platform used and conclusions from specific discoveries drawn, the connection between methodology and scientific soundness is missing.
1. Although this review paper categorizes main findings in one of the chapters 3.7. ST-derived Biomedical Advancements from several organ related carcinomas included in subchapters, the description of ST findings in tumor tissues in current format it is not acceptable.
2. Lack of understanding shown by authors of tumor microenvironment, tumor heterogeneity and cell-cell interactions all distinct processes, examined at the spatial transcriptomics, at the single cell sequencing level is a main critique and must be corrected prior publication.
3. Organ specific ST is summarized in superficial and main findings not categorized based on the method used, methods for data analysis used, and discoveries made all parts are not connected as they supposed to.
4. Tumor heterogeneity, does not belong under the same umbrella as cell-cell interactions, or Immune cell interaction, or microenvironment. Heterogeneity is a cancer cells property that allows cancer cells to be either dormant, aggressive or stem like, metastatic, or just actively proliferating present at the same time and space in the tumor tissue.
5. It is recommended to shorten this section to two three major examples, with in depth characterization of method and findings, by decreasing the number of organs and focusing on two three examples it is expected to improve the main message of this paper, how use of particular ST method and data deconvolution could help in major cancer biology pathways discovery.
6. Illustration of pathways deconvolution and cell identification might be more reasonable to be included in this in depth characterization of tumor biology or in context of pharmacological intervention.
7. It is recommended to shorten the 3.7. ST-derived Biomedical Advancements with examples of ST used in different malignancies to two/three examples from six currently under review, and build stronger section on data analysis, pathways mapping, cell identification and clustering into unique gene ontology clusters.
Comments on the Quality of English LanguageThe quality of English Language is fine, the text needs to be re-organized to be more focused on topic, succinct and precisely describing findings.
Round 2
Reviewer 2 Report
Comments and Suggestions for Authors
Authors corrected the paper sufficiently for its publication. Major critique was to shorten the paper, and give more in detail on molecular pathways that could be concluded from Spatial Transcriptomics.
It is advised to check for minor mistakes in the text, such as misspellings, grammar ect.
Author Response
We thank the Reviewer for their suggestions. All the changes made to the manuscript are marked with the "track changes" function, also including changes made during the first round of the revision.
Comment 1: It is advised to check for minor mistakes in the text, such as misspellings, grammar, etc.
As suggested, we have revised the whole manuscript and corrected minor language errors.